# Mattel’s Barbie: Investigation of a Symbol—Analysis of Polymeric Matrices and Degradation Phenomena for Sixteen Dolls from 1959 to 1976

**DOI:** 10.3390/polym14204287

**Published:** 2022-10-12

**Authors:** Andrea Macchia, Chiara Biribicchi, Camilla Zaratti, Katiuscia Testa Chiari, Martina D’Ambrosio, Denise Toscano, Francesca Caterina Izzo, Mauro Francesco La Russa

**Affiliations:** 1YOCOCU, Youth in Conservation of Cultural Heritage, Via T. Tasso 108, 00185 Rome, Italy; 2Department of Biology, Ecology and Earth Sciences DIBEST, University of Calabria, Via Pietro Bucci, Arcavacata, 87036 Rende, Italy; 3Department of Earth Sciences, Sapienza University of Rome, Piazzale Aldo Moro 5, 00185 Rome, Italy; 4Department of Environmental Biology, Sapienza University of Rome, Piazzale Aldo Moro 5, 00185 Rome, Italy; 5Interventi Conservativi Storico Artistici (ICSA s.r.l.), Via Oriano Sotto 2/A, 21018 Sesto Calende, Italy; 6Department of Environmental Sciences, Informatics and Statistics, Ca’ Foscari University of Venice, Via Torino 155, 30170 Venezia Mestre, Italy

**Keywords:** Barbie, PVC, plastic, degradation, plasticizer, EVA, conservation, ABS, LDPE, phthalates

## Abstract

Mattel’s Barbie dolls are the most famous and iconic dolls since 1959. Today, they are being collected by individuals and often conserved in museum environments due to their cultural and historical significance reflecting everyday life and historical events. However, just like most museum objects made of plastics, both historical and more recent Barbies show evident degradation phenomena. Firstly, Barbies were made of plasticized polyvinyl chloride (PVC), affected by the migration of additives—mostly the plasticizers—from the bulk phase to the outermost layer, appearing as a tacky and glossy exudate. Over the years, Barbies’ polymeric constituents were replaced with more stable ones, whose additives migration is limited compared to PVC, even though still occurring. Multispectral photography in visible (VIS) and ultraviolet (UV) light, microscopical observations in VIS and UV light, and Fourier Transform Infrared spectroscopy in the Attenuated Total Reflectance mode (FT-IR ATR) were performed to characterize the constituent materials of 15 Barbies produced between 1959 and 1976, bridging the information gap on their processing over the years. The micro-invasive multi-analytical approach also allowed for the characterization of the degradation products, permitting the reference of the exudated compound to the specific bulk polymer.

## 1. Introduction

Plastics entered the technological scene at the beginning of the 20th century. During the past decades, artists, designers, and architects have used these synthetic materials to create pieces that are now recognized as iconic items. Hence, they have become a significant part of many museum collections, posing major problems for collectors and conservators, due to their fast degradation [1]. Plastic items have defined our culture: ordinary objects, such as toys, are the reflection of significant social developments in political, cultural, scientific, and military history [2,3]. Children’s toys are part of social memory, mirroring the needs of the period when they were produced. It is possible to obtain information on the toy trade and understand the aspects of perception and use of historical culture [4].

Few toys have impacted fashion, style, and culture as intensely as Mattel’s Barbie doll, which was first produced in 1959 [5]. She is the best-selling and most famous doll ever [6]. Barbie has been the most iconic toy since the early 60s and continues to be very popular even today. It emphasized the importance of the female gender in society long before many living women had such opportunities [7]. Barbie was represented as a businesswoman as well as holding the usual gender roles for the 1960s [8]. Numerous versions of Barbie include various roles that a woman from the 1960s to the 1990s could hardly hope for, such as professor, librarian, nurse, vet, astronaut, NASCAR driver, babysitter, and presidential candidate [9]. This amazing fusion between Barbie and the community pushed the interest of museums in the preservation of Barbies [10].

From a material point of view, Barbies are plastic artifacts, prone to fast deterioration, and their conservation may represent a challenge as well as an opportunity for collectors and museum conservators. From a technological point of view, plastics are characterized by virtually endless possibilities for formulation of different polymers, with different characteristics [2,11]. Chemical-physical properties of polymers also depend on the presence of different additives, which are known to affect both the long-term stability and an object’s appearance during the degradation process [10,11,12]. One of the major problems associated with the applications of polymers is their instability in weathering and daylight [13,14]. It is well known that UV light and the short wavelengths of visible light (VIS) can induce photochemical processes in organic polymers, which result in dramatic changes in the polymeric matrix. The degradation phenomena can be measured in years, which is considerably less than that of traditional art and heritage materials, often measured in hundreds or thousands of years. Understanding the mechanisms of the reactions that produce degradation of polymer properties has thus been the subject of intensive research over the past 50 years [1,13,15,16,17,18]. In most cases, the main cause of deterioration is photooxidation, which is initiated by light irradiation [19]. Radiation in the ultraviolet and visible is indeed the primary factor in polymer weathering. Effects from heat, moisture, pollutants, mechanical stresses, and biological attacks also play a role, but in many cases, the weathering process begins with a photochemical event. Solar UV light induces photochemical processes in organic materials, which result in the fixation of oxygen, rearrangements, chain scissions, cross-linking, photo-discoloration, and photo-bleaching. All these processes contribute to a decrease in the various properties of the materials, such as optical, mechanical, and electrochemical properties [20].

Despite the very large effort put into the development of mechanical and chemical methods for assessing degradation, the links between chemical reactions and engineering failure in weathered polymers are far from being completely elucidated. In addition to sociological factors, an analysis of the evolution of Barbie dolls may include an assessment of their technological production, which can maintain industrial historical memory. Indeed, the Mattel doll underwent about 50,000 structural and design changes since its first release based on users’ needs and social changes [8,9,21]. Since the first Barbie, produced in 1959 using plasticized polyvinyl chloride (PVC), the constituent materials of Barbies have changed over time also because the bulk polymer was not stable enough, causing premature whitening due to the reaction of vinyl-based polymers with light [22]. The recipe of the polymer matrix changed both due to degradation problems and also because of economic and legal constraints of the specific production period [23].

To better understand the effect of these technological modifications, 15 Barbies produced between 1959 and 1976, based on the year reported on the Barbie itself, have been studied. The first aim of this study was to evaluate the Barbies’ constituent materials to bridge the information gap regarding the evolution of the most famous doll manufacturing process. Only a few data on Barbies’ processing changes over the years are available so far, making the selection of the most appropriate preservation strategy even more challenging [5,23]. Besides, each of them is characterized by different deterioration phenomena, mainly consisting of the formation of a glossy or crystal-like white patina on the surface which gives rise to dust embedding and color alteration over time. An effort has been made to relate constituent materials to specific degradation products using a micro-invasive analytical approach, permitting identification of compounds that tend to migrate to the surface layer. At the same time, the appearance of the different exudates was examined to attempt a prediction of their composition based on visual features.

## 2. Materials and Methods

Fifteen Barbie dolls have been analyzed. Their dating is as follows:1959/1960: Barbies #1 and 2.1964: Barbies #3 to 6.1965: Barbies #7 and 8.1972: Barbies #9 and 10.1974/1976: Barbies #11 to 15.

The Barbies’ general conditions were examined through macroscopic and microscopic inspection. Multispectral photography was carried out using a multispectral system (Madatec, Milan – Italy), consisting of a Digital Camera (Samsung NX500—28.2 MP BSI CMOS) and Madatec spotlights with 365 nm wavelength (UV light). Images of the induced fluorescence were taken using the HOYA UV-IR filter cut 52 (Kenko Tokina Co. Ltd., Tokyu – Japan) [24,25,26]. An optical portable microscope Dino-Lite AM411-FVW (Dino-Lite Europe, Almere – Netherlands) was used at different magnifications—from 40× to 220×—and illumination modes—visible light (VIS), ranking light (VIS-RAD), and ultraviolet light (UV) to better evaluate the deterioration phenomena on the surface of the polymeric matrix. The acquired images enabled the selection of the most significant areas for the subsequent analysis performed using Fourier Transform Infrared spectroscopy in the Attenuated Total Reflectance mode (FT-IR ATR). Most of the FT-IR ATR analysis was carried out by directly placing each element comprising the dolls on the diamond of the ATR, working in a non-invasive mode. Based on the details acquired using the portable microscope and imaging investigations with UV-induced fluorescence, a few deterioration products were sampled using a scalpel or a swab with absolute ethanol solution and analyzed through FT-IR ATR to better understand the composition of the deteriorated areas. FT-IR ATR spectra were collected using the Nicolet Summit FT-IR spectrometer and the Everest™ Diamond ATR accessory (ThermoFisher Scientific™, Waltham, MA, USA) with a resolution of 8 cm^−1^ and a laser frequency of 15,798.0 cm^−1^. 32 scans were performed on each area. The acquired spectra were examined using the instrument library, scientific literature, and the database edited by Vahur et al. [27,28].

## 3. Results

### 3.1. Imaging in Visible (VIS) Light and Ultraviolet-Induced Fluorescence (UV)

Images in visible light and UV-induced fluorescence acquired on the different Barbies (front and back) allowed for the identification of the constituent materials. The analysis also enabled the collection of significant information on the Barbies’ state of conservation based on the different fluorescence emissions. The analyzed Barbies showed different fluorescence effects depending on the examined body part (pink, orange, and dark emissions). The main differences can be seen in the legs, arms, and torso (Table 1 and Table 2).

Despite the year of production, the hair of each brunette Barbie exhibits strong absorption of the UV radiation, while the hair of blonde Barbies has different degrees of yellow fluorescence emission based on the color. Only the hair of Barbie #13 shows yellow-greenish emission.

The Barbies #1, 2, 3, and 4 exhibit the typical dim green fluorescence emission of the polyvinyl chloride (PVC) polymer [29]. The Barbie #4 is characterized by the presence of a substance on the torso, showing a white color in visible light (VIS), while a darker fluorescence emission under UV light if compared to the other parts of the body. The crystal-like white exudate is also present on the legs of Barbie #5 and the face of Barbie #6, showing emissions lighter than in the legs and the face, respectively. It is possible to observe a UV radiation-absorbing material on the torso of Barbies #6, 7, 8, 9, 10, 11, and 12, appearing deep dark under UV light compared to the face and limbs, which glow intensively when exposed to UV radiation. In addition, the face and limbs of Barbies #13, 14, and 15 exhibit high absorption of the UV radiation appearing dark if compared to the torsos. Even though the difference cannot be seen under UV light, the faces of Barbies #1, 4, 6, 7, 8, 9, 10, 11, 12, 13, and 14 appear darker if compared with other parts of the body with the same fluorescence emission, due to the presence of an exudate that makes the surface sticky and embedded with dust over time (torso, face, and legs of #1, torso and arms of #4, legs and face of #6, legs of #7, face and legs of #8, limbs of #9, face and legs of #10, limbs of #11, legs of #12, and arms and legs of #13 and #14). Barbies #9, 10, 11, and 13 show differentiated fluorescence emission based on the examined area, even though the difference is not noticeable in VIS light. Indeed, only Barbie #10 exhibits dark-reddish spots—in VIS light—on the upper part of the legs, appearing pink-fluorescent when exposed to UV radiation. On the other hand, it is possible to observe multiple fluorescence effects on Barbies #9, 11, and 12, that do not appear deteriorated in VIS light. Barbies #9 and 11 are characterized by a pinkish fluorescence on the inner and the upper sides of the legs, while strong absorbance of the UV radiation—resulting in a dark appearance—is visible on the remaining parts of the legs, as well as on the torso and arms. This phenomenon is especially evident in Barbie #11, which shows an unevenly spread substance on the arms, appearing deep dark under UV light. In both cases, the face of the doll shows a yellow-greenish fluorescence emission. Barbie #13 exhibits a pinkish fluorescence on the calves and a darker one on the limbs, on which attached fragments of her hair are visible.

The dark-absorbing material that is present on the limbs of most of the Barbies is characterized by different degrees of saturation. This effect could be most likely related to different amounts of plasticizer on the surface. Indeed, migration of the plasticizer is a common deterioration problem in plastics, causing embrittlement of the bulk and adhesion of environmental pollutants and dust on the object’s tacky surface, altering its color and fluorescence emission [22]. In other cases, the differences visible under UV radiation could be related to the constituent polymers, which changed over the years.

The parts of the Barbies exhibiting different fluorescence responses are as follows:The face in Barbies #14 and 15.The limbs in Barbies #5, 7, 9, 11, 12, 13, 14, and 15.The torsos of Barbies #4, 5, 6, 7, 8, 9, 10, 11, and 12.

### 3.2. Microscopic Observations

Microscopic acquisitions were taken using a 50× magnification to further examine the morphology and the state of conservation of the 15 Barbies (Table 3). The analysis confirms the hypotheses made using multispectral imaging. Faces and legs appear to be the most degraded areas. Indeed, they are often characterized by the presence of an exudate that makes the surface sticky and glossy. This substance is particularly visible on the arms and legs of Barbie #4, but also on the face of Barbie #7—where embedded fibers can be seen—and on the face and limbs of Barbies #8 and #9. Other deterioration phenomena are also better highlighted, such as the white crystal-like substance on the back of Barbie #4, on the legs of Barbie #5, and on the face of Barbie #6. Some of the Barbies also present materials superimposed by the owners, such as the red lipstick on Barbies #1 and 2, scratches induced by scrubbing on Barbie #7, cracking in Barbie #9, and signs of a marker pen ink in Barbies #10, 12, 13, and 15. These alterations of the original doll were not further investigated in this study, as they are considered accidental or intentional damages that cannot be related to intrinsic deterioration phenomena occurring in plastic materials.

### 3.3. Fourier Transform Infrared Spectroscopy in the Attenuated Total Reflectance Mode (FT-IR ATR)

Spectra were collected examining each element comprising the dolls to characterize their constituent materials and the substances responsible for the evident degradation of the Barbies.

Based on their constituent materials, the Barbies were divided into three groups, namely:Group I: Barbies #1, 2, 3, 4, 5;Group II: Barbies #6, 7, 8, 9, 10, 11, 12;Group III: Barbies #13, 14, 15.

Figure 1 depicts an FT-IR ATR spectrum representative for Group I: the torso, limbs, and face of Barbies of this Group consist of a PVC matrix, which was identified thanks to the characteristic peaks at 1426 cm^−1^ (CH_2_–Cl angular deformation), 960 cm^−1^ (C–H out of plane trans deformation), 831, 691, and 615 cm^−1^ (C–Cl bond stretching) [30]. A peak at 1640 cm^−1^ was detected in almost all the spectra acquired on the body of the five Barbies, except for the torso of doll #4 and the legs of #5. This absorption can refer to the presence of carboxylates in the solid products of degradation of PVC and be attributed to the formation of double bonds in the main polymer chain, indicating that some degradation occurred during the crosslinking process of PVC [31]. Spectra acquired on the limbs, torso, and face of the Barbies of Group I—except for the torso of #4—highlighted the presence of the sole plasticizer, identified as a phthalate-based compound for the bands at 1718 cm^−1^ (phthalate ester), 1450 cm^−1^ (C–H bending), 1278 cm^−1^, 1127, and 1027 cm^−1^ (O–CH_2_ aromatic group stretching), the small peaks at 1579 cm^−1^ and 1599 cm^−1^ (aromatic ring quadrant stretching vibration), and the strong absorbance band at 743 cm^−1^ (phthalate ortho-substituted aromatic ring) [22,32,33]. It is likely that this can be linked to the presence of a dioctyl phthalate (DOP) or di(2-ethylhexyl) phthalate (DEHP) that have been traditionally used as plasticizers in PVC objects (Figure 1) [34,35,36].

The hair of Barbies belonging to Group I has been identified as polyvinylidene dichloride (PVDC) for the peak at 1405 cm^−1^ (CH_2_ bending), the two peaks at 1068 and 1044 cm^−1^ (C–O stretching vibrations), and the band a 655 cm^−1^ (C–Cl_2_ stretching vibrations), while the small peak at 1740 cm^−1^ corresponding to the C=O stretching is linked to the presence of a plasticizer (Figure 2) [37].

Barbie #4 shows a white crystal-like patina on the torso. The FT-IR ATR analysis detected di(1-butyl)tin(IV) phthalate, highlighting the presence of organotin compounds and plasticizers in the polymeric matrix (Figure 3). Organotin compounds have been commonly used as a thermal stabilizer for PVC compounds even before 1973 due to PVC’s sensitivity to thermal degradation during processing and use [38]. Phthalate-based plasticizers have been also widely added to improve the flexibility, processability, and durability of PVC [39]. Both organotin compounds and plasticizers tend to migrate to the surface of the polymer because they are usually not chemically bonded to the polymer chain [40,41]. Release of the plasticizer is one of the main causes of the deterioration of plasticized PVC, resulting in mass loss, embrittlement of the bulk, and the appearance of sticky exudates on the surface, which could appear as white deposits on the torso of Barbie #4 due to the presence of organotin compounds.

The torso of Barbies from Group II consists of low-density polyethylene (LDPE) matrix (Figure 4). LDPE can be distinguished from high-density polyethylene (HDPE) by the small peak at 1377 cm^−1^ [42]. The polymer can be also characterized by the peaks at 1467 cm^−1^ (CH_2_ bending vibrations), 730 and 717 cm^−1^ (CH_2_ rocking vibrations), while the small peak at 1720 cm^−1^ has to be related to the presence of phthalate plasticizers, even if in smaller amounts than in the PVC parts. The hair of Barbies belonging to Group II again consists of polyvinylidene dichloride (PVDC). Limbs and heads are made of PVC and show the characteristic peaks of the phthalate plasticizer—presumably DEHP or DOP—in the yellow-green-fluorescing areas as well as in the pink-fluorescing ones.

The torso of Barbies from Group III has been characterized as acrylonitrile butadiene styrene (ABS), which explains the yellow-orange fluorescence under UV light compared to the other dolls (Figure 5). ABS’ characteristic peaks can be seen at 1726 cm^−1^ (C=O stretching), 1583 cm^−1^ (aromatic stretching vibration), 1492 cm^−1^ (aromatic C=C bond), 1452 cm^−1^ (C–H bending), 1070 and 1028 cm^−1^ (C–O stretching), 965, 910, 758, and 698 cm^−1^ (C=C bending) [43]. The leg and face of all the Barbies of Group III consist of PVC, while differences can be seen in the spectra collected on the hair and the face. Indeed, while the arms of Barbie #13 are made from PVC, the ones of Barbie #14 show the typical peaks of LDPE.

Furthermore, the spectra acquired from the arms of Barbie #15 can be attributed to ethylene vinyl acetate (EVA) with peaks at 724 cm^−1^ (CH_2_ rocking vibrations), 1027 and 1242 cm^−1^ (C–O stretching), 1380 and 1460 cm^−1^ (C–H bending), and 1737 cm^−1^ (C=O stretching) (Figure 6) [44].

Finally, Barbie #14 and 15 again have PVDC hair, whereas Barbie #13 has polypropylene (PP) hair, which is the reason for the green fluorescence emission when irradiated by UV light, as reported in paragraph 3.1 [45]. Characteristic peaks of PP can be observed at 2950, 2920, and 2870 cm^−1^ (C–H stretching), 1456 cm^−1^ (C–H bending), 1376 cm^−1^ (O–H bending), 1166 cm^−1^ (C=O stretching), 996, 973, 840, and 808 cm^−1^ (C=C bending) (Figure 7) [46].

A summary of the polymers composing each element of the 15 examined Barbies is reported in Table 4.

## 4. Discussion

The results obtained by the multi-analytical approach highlighted the frequent use of PVC in Barbies’ manufacturing from 1959 to 1965, which has been gradually replaced by more stable polymers over the years. The characterization of the different plastic elements and degradation products was made possible by using FT-IR ATR spectroscopy, which confirmed the differences in the fluorescence emissions of the plastic elements comprising the 15 Barbies. Indeed, multispectral images in UV light show PVC’s yellowish-green fluorescence emission, while LDPE, which was used to produce the torsos between 1964 and 1974/1976, deeply absorbs the UV radiation, resulting in a dark appearance. Finally, polymers used since 1974/1976—ABS, PP, and EVA—result in yellow-orange, glow-green, and yellow fluorescence emissions, respectively, when irradiated by UV light sources. Barbies showing a grey patina or a pink-reddish fluorescence under UV light—meaning the PVC elements mainly—are the ones that feature the highest inner alteration and the greatest amount of plasticizer on the surface. The presence of phthalate-based plasticizers was evident both in the FT-IR ATR spectra acquired on samples and the ones resulting from the direct analysis of PVC dolls. The doll exhibiting the worst state of conservation is Barbie #11, which has different fluorescence colors on various elements of her body. The FT-IR spectra also confirmed the hypothesis that the main cause of degradation, referring to the glossy exudate and the white crystal-like patina, is the migration of the plasticizers—phthalates—and stabilizers—such as organotin compounds—associated with the polymer processing to improve its flexibility and heat stability.

## 5. Conclusions and Further Research

This study has provided further information regarding the evolution of the Barbies’ manufacturing process over the years, highlighting changes in the choice of the polymeric matrix for the different body elements. The analytical results were associated with the visible deterioration phenomena that were present on all the Barbies’ surfaces to different degrees. The migration of phthalate-based plasticizers from the bulk up to the surface of the polymer can be considered the main cause of the deterioration of the analyzed dolls. Such a phenomenon results in physical damage to the bulk—embrittlement—and the formation of a tacky patina on the PVC’s outermost layer, which was evident in both VIS and UV light when using the imaging system and portable microscopy.

It was likely for this reason that the Bakelite company carried out an extensive investigations back in 1954 [22,47]. Based on the results from analysis and previous reports, the producers of Barbie dolls had to gradually rethink the chemical composition even before the late 1980s, when a German government law limited the amount of plasticizer allowed in PVC for the safety of children playing with PVC toys [5,23]. Mattel eventually started adapting Barbies’ constituent materials and packaging to the law up to the present day [23]. Over the years, Barbie dolls started to have EVA arms, ABS torsos, heads of a hard vinyl compound, and outer legs in PVC, although using a different formula from the earlier dolls [23]. Even though the evolution process is known to have occurred, only a little information is available regarding the dates on which these structural changes were made. Thus, the present research represents an unprecedented deeper view of the gradual evolution of Barbies’ manufacturing, which enabled assessing the introduction of new polymers—such as LDPE—already used in 1964, and newer ones—such as ABS, PP, and EVA—evident in 1974-1976. Some of these materials are still in use in Mattel’s industry, suggesting that significant changes in the Barbies’ chemical composition occurred already around the ‘70s. Eventually, the implemented multi-analytical approach also allowed for the understanding of polymers’ tendency to deterioration phenomena. Degradation—meaning the migration of the plasticizer and other additives on the outermost surface layer—appears to be more consistent in PVC: the other polymers show plasticizer migration as well, but to a lesser degree.

Nevertheless, given the inner instability of the detected polymeric materials, future research will be devoted to a more in-depth investigation of polymeric fractions, additives, and their degradation products and sub-products (e.g., by micro-invasive analysis by Py-GC-MS or TG-DSC). Then, special attention will be paid to the definition of an appropriate conservation strategy to prevent future deterioration caused by new migration of the additives to the surface. The rapid deterioration of these plastic dolls represents an urgent concern for Cultural Heritage conservation. Hence, innovative preservation approaches need to be improved and tested.

## Figures and Tables

**Figure 1 polymers-14-04287-f001:**
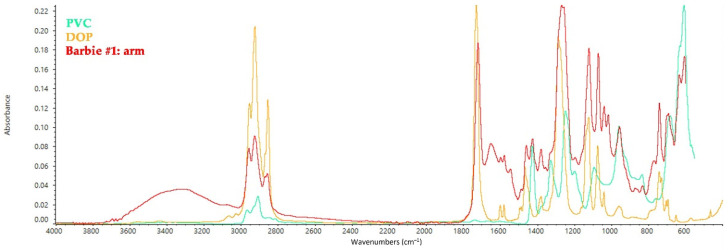
FT-IR ATR spectrum collected on the arm of Barbie #1 and shown as a model spectrum of PVC elements in all the other dolls, with the reference spectra of PVC and n-octyl n-decyl phthalate (plasticizer).

**Figure 2 polymers-14-04287-f002:**
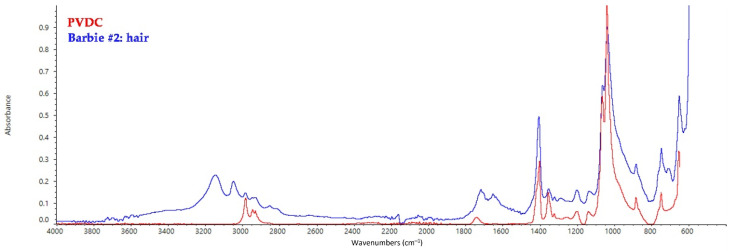
FT-IR ATR spectrum collected on the hair of Barbie #2 and shown as a model spectrum of PVDC hair in all the other dolls, with the reference spectrum of PVDC.

**Figure 3 polymers-14-04287-f003:**
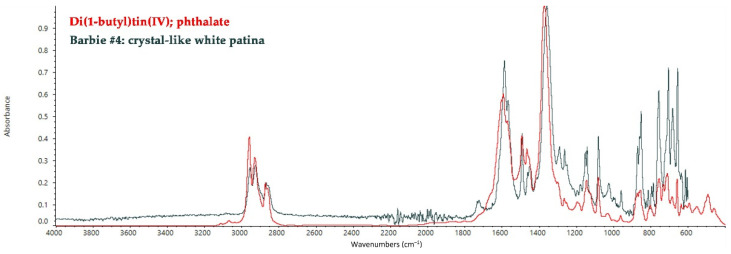
FT-IR ATR spectrum collected on the crystal-like white patina on the torso of Barbie #4, with the reference spectrum of a phthalate-organotin compound.

**Figure 4 polymers-14-04287-f004:**
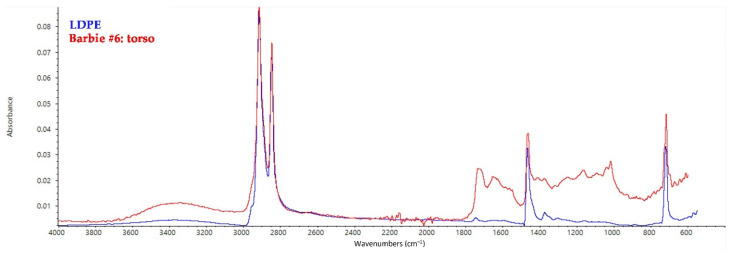
FT-IR ATR spectrum collected on the torso of Barbie #6 and shown as a model spectrum of LDPE elements in all the other dolls, with the reference spectrum of LDPE.

**Figure 5 polymers-14-04287-f005:**
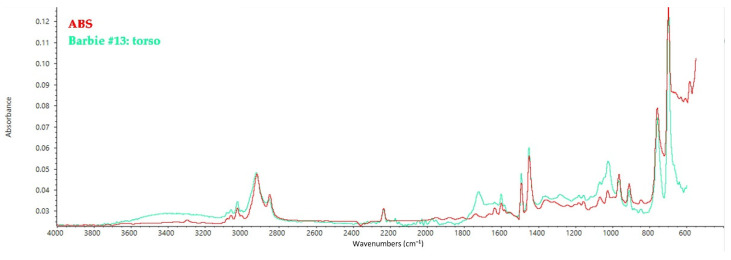
FT-IR ATR spectrum collected on the torso of Barbie #13 and shown as a model spectrum of ABS elements in all the other dolls, with the reference spectrum of ABS.

**Figure 6 polymers-14-04287-f006:**
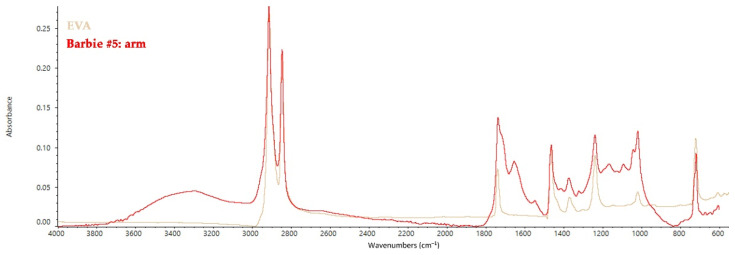
FT-IR ATR spectrum collected on the arm of Barbie #15 with the reference spectrum of EVA.

**Figure 7 polymers-14-04287-f007:**
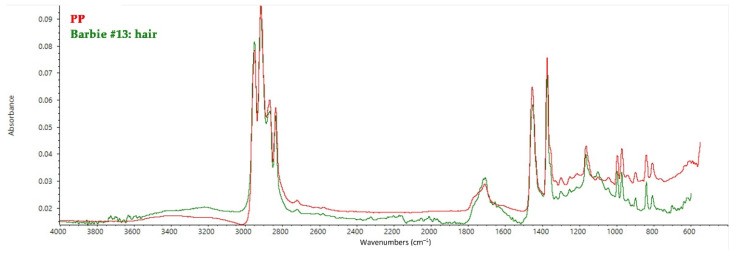
FT-IR ATR spectrum collected on the hair of Barbie #13 with the reference spectrum of PP.

**Table 1 polymers-14-04287-t001:** Multispectral images (VIS and UV) of the 15 Barbies including a VIS/UV reference standard: right side.

**Barbie #1 (1959/60)**		**Barbie #2 (1959/60)**		**Barbie #3 (1964)**	
VIS	UV	VIS	UV	VIS	UV
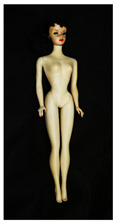	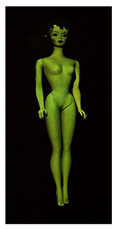	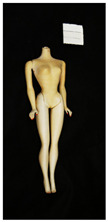	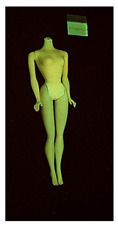	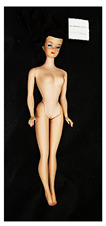	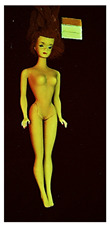
**Barbie #4 (1964)**		**Barbie #5 (1964)**		**Barbie #6 (1964)**	
VIS	UV	VIS	UV	VIS	UV
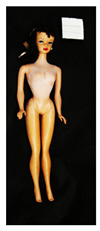	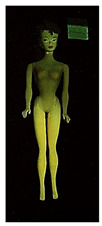	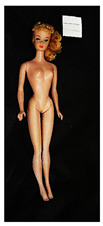	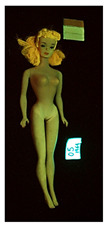	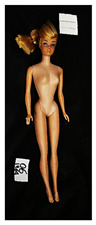	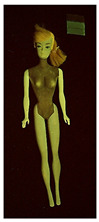
**Barbie #7 (1965)**		**Barbie #8 (1965)**		**Barbie #9 (1972)**	
VIS	UV	VIS	UV	VIS	UV
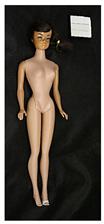	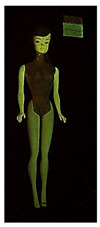	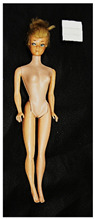	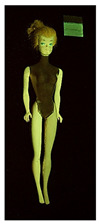	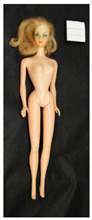	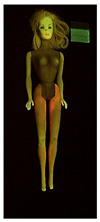
**Barbie #10 (1972)**		**Barbie #11 (1974/76)**		**Barbie #12 (1974/76)**	
VIS	UV	VIS	UV	VIS	UV
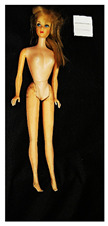	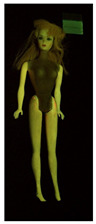	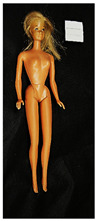	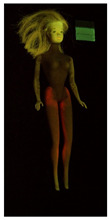	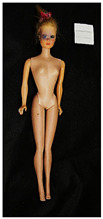	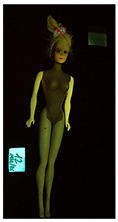
**Barbie #13 (1974/76)**		**Barbie #14 (1974/76)**		**Barbie #15 (1974/76)**	
VIS	UV	VIS	UV	VIS	UV
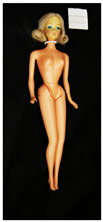	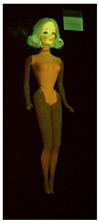	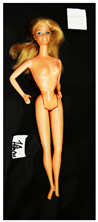	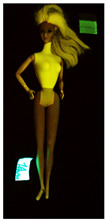	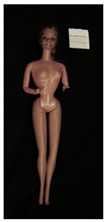	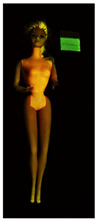

**Table 2 polymers-14-04287-t002:** Multispectral images (VIS and UV) of the 15 Barbies including a VIS/UV reference standard: backside.

**Barbie #1 (1959/60)**		**Barbie #2 (1959/60)**		**Barbie #3 (1964)**	
VIS	UV	VIS	UV	VIS	UV
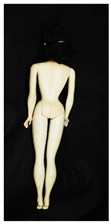	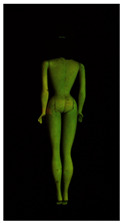	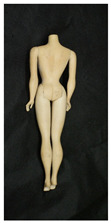	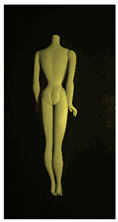	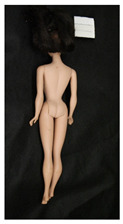	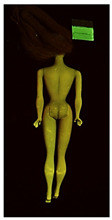
**Barbie #4 (1964)**		**Barbie #5 (1964)**		**Barbie #6 (1964)**	
VIS	UV	VIS	UV	VIS	UV
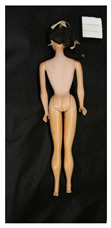	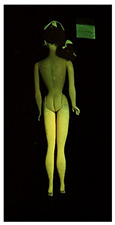	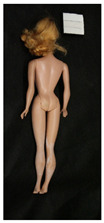	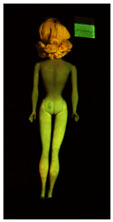	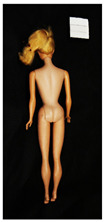	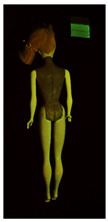
**Barbie #7 (1965)**		**Barbie #8 (1965)**		**Barbie #9 (1972)**	
VIS	UV	VIS	UV	VIS	UV
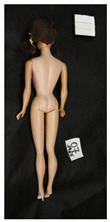	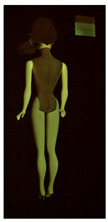	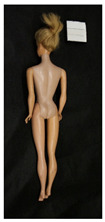	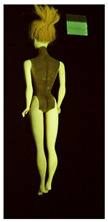	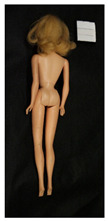	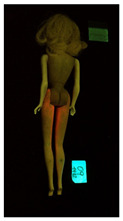
**Barbie #10 (1972)**		**Barbie #11 (1974/76)**		**Barbie #12 (1974/76)**	
VIS	UV	VIS	UV	VIS	UV
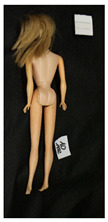	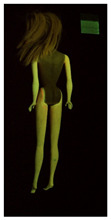	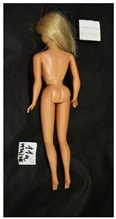	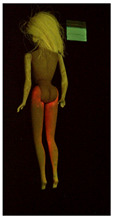	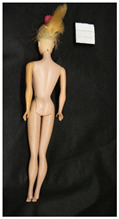	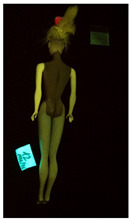
**Barbie #13 (1974/76)**		**Barbie #14 (1974/76)**		**Barbie #15 (1974/76)**	
VIS	UV	VIS	UV	VIS	UV
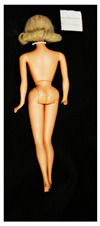	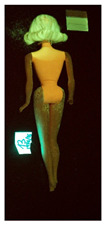	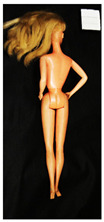	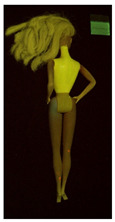	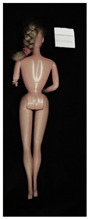	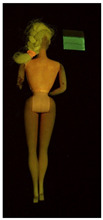

**Table 3 polymers-14-04287-t003:** Microscopic images (magnification: 50×) under VIS and UV light: details of the head, torso, legs, arms, and back on each Barbie identified through the ID number.

ID	Head	Torso	Legs	Arms	Back
1	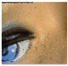	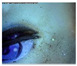	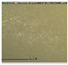	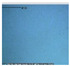	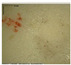	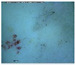	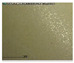	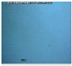	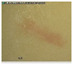	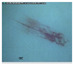
2	NA	NA		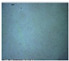	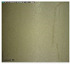	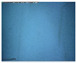	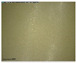	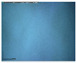	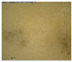	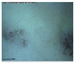
3	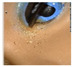	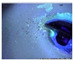	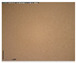	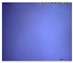	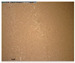	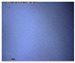	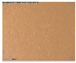		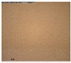	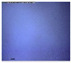
4	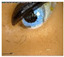	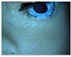	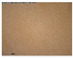	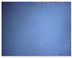	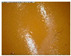	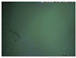	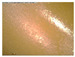	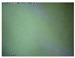	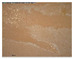	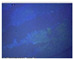
5	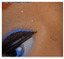	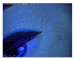	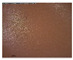	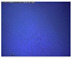	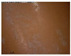	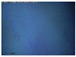	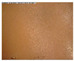	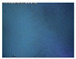	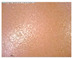	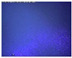
6	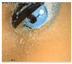	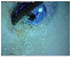	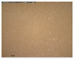	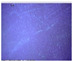	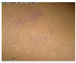	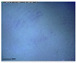	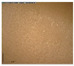	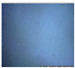	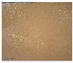	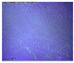
7	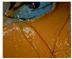	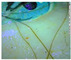	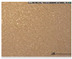	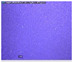	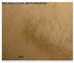	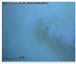	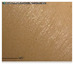	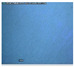	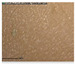	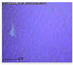
8	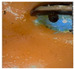	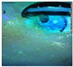	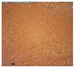		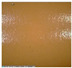	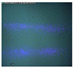	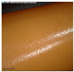	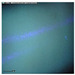	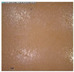	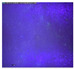
9	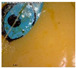	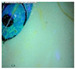	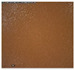	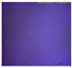	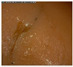	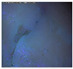	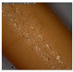	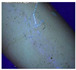	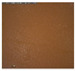	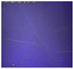
10	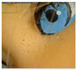	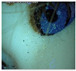	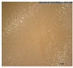	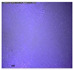	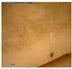	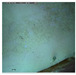	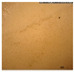	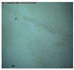	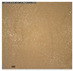	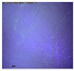
11	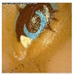	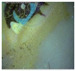	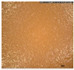	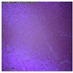	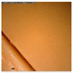	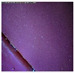	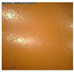	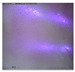	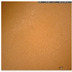	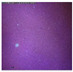
12	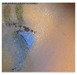	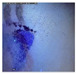	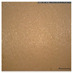	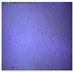	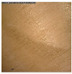	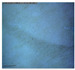	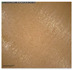	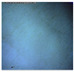	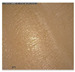	
13	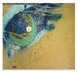	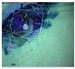	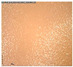	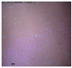	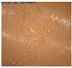	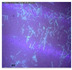	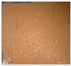	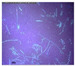	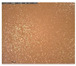	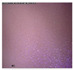
14	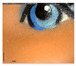	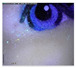	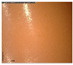	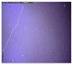	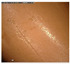	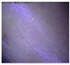	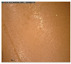	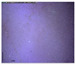	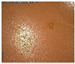	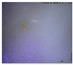
15	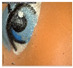	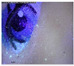	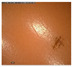	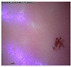	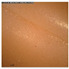	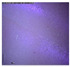	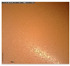	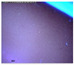	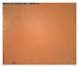	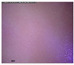

**Table 4 polymers-14-04287-t004:** Results of the FT-IR ATR analysis, based on the characteristic peaks of the Barbies’ constituent materials.

ID	Year	Torso	Arms	Legs	Face	Hair
1	1959/1960	PVC	PVC	PVC	PVC	PVDC
2	1959/1960	PVC	PVC	PVC	PVC	PVDC
3	1964	PVC	PVC	PVC	PVC	PVDC
4	1964	PVC	PVC	PVC	PVC	PVDC
5	1964	PVC	PVC	PVC	PVC	PVDC
6	1964	LDPE	PVC	PVC	PVC	PVDC
7	1965	LDPE	PVC	PVC	PVC	PVDC
8	1965	LDPE	PVC	PVC	PVC	PVDC
9	1972	LDPE	PVC	PVC	PVC	PVDC
10	1972	LDPE	PVC	PVC	PVC	PVDC
11	1974/1976	LDPE	PVC	PVC	PVC	PVDC
12	1974/1976	LDPE	PVC	PVC	PVC	PVDC
13	1974/1976	ABS	PVC	PVC	PVC	PP
14	1974/1976	ABS	LDPE	PVC	PVC	PVDC
15	1974/1976	ABS	PEVA	PVC	PVC	PVDC

## Data Availability

Publicly available datasets were analyzed in this study to compare the acquired FT-IR spectra with already available spectra of standard materials. Data can be found here: https://spectra.chem.ut.ee/ (accessed on 5 September 2022).

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
