# Peer review of "Mattel’s Barbie: Investigation of a Symbol—Analysis of Polymeric Matrices and Degradation Phenomena for Sixteen Dolls from 1959 to 1976"

_polymers, 2022, doi:10.3390/polym14204287_

Round 1

Reviewer 1 Report

This manuscript reports the results of an investigation of the composition and degradation of materials used in Barbie doll construction. Much of it reflects the migration of plasticizer from PVC, a material utilized for construction of early dolls. In many ways, this represents a somewhat unusual choice of research topic and the results might be as well suited for publication in a sociology or museum preservation journal as in Polymers.

The manuscript will require some revision for clarity and readability. Corrections are penciled-in directly on pages of the manuscript attached. These are illustrative of the kinds of changes needed throughout. In revision, careful attention should be paid to the use of tenses and proper sentence structure. Awkward phrases should be avoided - for example, "results of the analysis and literature" should be, "results from the analysis and previous reports." Personal pronouns should be omitted. 

Author Response

Thank you for your comments and suggestions. We improved the readability of the manuscript based on your corrections.

Reviewer 2 Report

I found this to be an interesting topic and the authors do a good job describing this issue, their approach and results. Overall, while this manuscript is definitely atypical I believe it has merit and will be of use/interest to polymer scientists/engineers. Some minor comments to be addressed:

1 -On page 7 line 167 the authors note "surface sticky in some of them." This should/could be more specific especially as this observation of stickiness is returned to later in the discussion.

2 - line 222 has a typo "Gropu"

3- line 228 should be edited " degradation of PVC And an be attributed..."

4-The manuscript results focuses on the FTIR analysis while it is generally okay, it could be better presented. For instance baseline correction and perhaps normalizing to a known base polymer peak may allow for better visual comparison. This would be very easy to do and help with the comparison of the respective peak assignments.

5-The authors note on page 350 how further analysis will be done on the degradation products, though I was wishing that was included here. Similar for the expressed desire to provide conservation strategies for this identified issue with these polymers and the additives described.

Author Response

Review 2

I found this to be an interesting topic and the authors do a good job describing this issue, their approach and results. Overall, while this manuscript is definitely atypical I believe it has merit and will be of use/interest to polymer scientists/engineers.

Answer

Thank you for your comments and suggestions. We improved the readability of the manuscript based on your corrections.

Some minor comments to be addressed:

Comment

1 -On page 7 line 167 the authors note "surface sticky in some of them." This should/could be more specific especially as this observation of stickiness is returned to later in the discussion.

Answer

Solved. We added specifics on the Barbies’ constituent elements.

Comment

2 - line 222 has a typo "Gropu"

Answer

Solved.

Comment

3- line 228 should be edited " degradation of PVC And an be attributed..."

Answer

Solved.

Comment

4-The manuscript results focuses on the FTIR analysis while it is generally okay, it could be better presented. For instance baseline correction and perhaps normalizing to a known base polymer peak may allow for better visual comparison. This would be very easy to do and help with the comparison of the respective peak assignments.

Answer

Thank you for the suggestion. We used the baseline correction improving the visualization of the spectra. Due to the difference in the peaks’ intensity among different polymers, we believe that visual comparison would not benefit from normalization to a known base polymer peak in this case.

Comment

5-The authors note on page 350 how further analysis will be done on the degradation products, though I was wishing that was included here. Similar for the expressed desire to provide conservation strategies for this identified issue with these polymers and the additives described.

Answer

Thank you for your suggestion. Even though including further analyses of the degradation products would improve the significance of the presented results, they have yet to be performed, as well as the definition of a conservation strategy. We believe that such topics should be included in other research papers.

Reviewer 3 Report

Thank you for giving me the opportunity to review this manuscript.

The article isonly based on the basic material analysis using spectral techniques. Apart from conclusions on the indication of when and  what material was used to produce individual doll fragments, there is no scientific novelty. The migration of plasticizers, as mentioned by the authors, is also obvious and known. If there were at least a discussion about limiting further degradation of these valuable exhibits, the article would gain in scientific value.

Nevertheless, the article is interesting and I read it with pleasure, because who did not meet Barbie in childhood ?! As the authors wrote, it is an icon, and the article should be published as an interesting review article on the evolution of the use of polymeric materials in the production of toys.

Please note the spelling of PVC – should be written poly(vinyl chloride).

Author Response

Thank you for your positive comments.

Reviewer 4 Report

The manuscript submitted by Andrea Macchia et al reported on the “Mattel’s Barbie: investigation of a symbol. Analysis of poly-meric matrices and degradation phenomena of sixteen dolls from 1959 to 1976” In this work, different barbie dolls were examined carefully in terms of degradation and many characterizations were performed such as UV, Vis and FTIR.

Although the paper is not hitting on the hot area of polymer research, it is really interesting and amazing that the authors focus on degradation of toys. I believe this manuscript was well organized and professional. Even though not much experiments are displayed, I still prefer to accept this article since it contributes to a special research area which is always being ignored.

The introduction shows about the history of barbie manufacturing and the polymer side behind it. The UV-Vis experiments are in details and the discussion are fruitful. I can get the dedication the authors put into this topic though it is far away from the hot polymer area. I really like the topic and the passion behind it.

I suggest that it can be accepted without modification.

Author Response

We really appreciate the reviewer's comment on our manuscript.